# Stress Rupture Life Prediction Method for Notched Specimens Based on Minimum Average Von Mises Equivalent Stress

**Dawei Ji [1,2], Xianming Hu [1,2], Zuopeng Zhao [1,2], Xu Jia [1,2], Xuteng Hu [1,2,*] and Yingdong Song [1,2,3]**

[1]  College of Energy & Power Engineering, Nanjing University of Aeronautics and Astronautics, Nanjing 210016, China; jidawei@nuaa.edu.cn (D.J.); 15950537166@163.com (X.H.); zpzhao@nuaa.edu.cn (Z.Z.); xujiacepe@nuaa.edu.cn (X.J.); ydsong@nuaa.edu.cn (Y.S.)

[2]  Key Laboratory of Aero Engine Thermal Environment and Thermal Structure, Ministry of Industry and Information Technology, Nanjing University of Aeronautics and Astronautics, Nanjing 210016, China

[3]  State Key Laboratory of Mechanics and Control of Mechanical Structures, Nanjing University of Aeronautics and Astronautics, Nanjing 210016, China

*  Correspondence: xthu@nuaa.edu.cn

**Abstract:** Creep tests were carried out on notched plate specimens of nickel-based superalloy GH4169 with different stress concentration coefficients. It was found that the duration of the first stage of the creep curve increases with the increase of stress concentration coefficient, while the fracture ductility decreases with the increase of stress concentration coefficient. To predict the life of notched plate specimens, four constitutive models were used to analyze the stress and strain of the notches. It was found that the average Von Mises equivalent stress (AVES) on the minimum notch section first decreases and then increases with the creep time, resulting in a minimum value. The minimum average Von Mises equivalent stress (MAVES) is considered as the characteristic stress of notched specimens in this paper. The creep life equation is fitted according to the results of creep tests of smooth specimens, and then the predicted life of notched specimens is obtained by substituting the minimum average Von Mises equivalent stress of notched specimens into the creep equation. The prediction results of the four constitutive models are within 2 times the dispersion band, and the three-stage model is within the 1.5 times dispersion band.

**Keywords:** nickel-based superalloy; creep rupture; life prediction; notched specimens; average Von Mises equivalent stress

## 1. Introduction

The hot section components of aero-engine often operate at relatively high temperatures for a prolonged time. The complex loads and irregular shapes result in a multiaxial state of stress in the components. To study the creep behavior of this kind of component in the complex working environment and predict their life, the first step is to accurately predict the stress rupture life of notched specimens under a laboratory environment.

The research on the rupture life of notched specimens at high temperatures has gradually increased since the 1970s, among which the notched round bar specimens are widely researched. The commonly used methods include the skeletal point method [1–6], the multi-axial ductility exhaustion method [7–11], and the prediction method based on AVES [12]. These methods cannot be operated without the finite element analysis of notched specimens. Hayhurst [6] et al. used skeletal point stress for the prediction of rupture life. They found that in the process of stress redistribution, there was a point at the minimum cross-section where the stresses are nearly constant. The skeletal point stresses have been used to characterize the deformation and failure behavior of the material under multiaxial creep conditions. This model is suitable for notched round bar specimens to find the skeletal point along the radius direction. While if the cross-section of the specimen is rectangular or irregular like that in the real components, it is not easy to find the skeletal point. The

multi-axial ductility exhaustion approach establishes the relationship between uniaxial and multiaxial creep ductility through the multiaxial ductility factor, which is generally a function of hydrostatic pressure. Webster and Pickard [12] performed finite element analysis on the notched specimens and obtained the AVESes at a certain time. Using the concept of critical rupture ductility (i.e., creep failure through ductility exhaustion), strain accumulation is assumed to follow the steady-state rate consistent with the AVES until the rupture ductility is reached.

It is necessary to carry out a large number of long-term creep experiments to obtain the creep ductility curve. This paper proposes a new method for predicting rupture life based on AVESes, without using the concept of critical rupture ductility. According to our research, it is found that the AVES at the minimum cross-section of notched specimens first decreases and then increases with time, so there exists a MAVES. The MAVES is taken as the characteristic stress and is substituted into the creep life equation to obtain the rupture life of notched specimens. The influence of four different creep constitutive equations and three different creep life equations on the life prediction is analyzed in this paper.

## 2. Experiment and Results

### 2.1. Composition and Microstructure Characterization

GH4169 is a precipitation strengthening nickel-based superalloy with good fatigue and corrosion resistance. It is widely used in aero-engine rotor components, such as high-pressure compressor disks and turbine disks. The composition of GH4169 is similar to that of the nickel-based superalloy IN718 [13], as shown in Table 1.

**Table 1.** Composition of Ni-based superalloy GH4169.

| Element | Cr | Ti | Fe | Mo | Al | Nb |
|---|---|---|---|---|---|---|
| Wt% | 17.00~21.00 | 0.75~1.15 | 14.2~24.0 | 2.80~3.30 | 0.30~0.70 | 5.00~5.50 |
| Element | N | C | Mn | Si | P | Ni |
| Wt% | ≤0.01 | 0.015~0.006 | ≤0.35 | ≤0.35 | ≤0.015 | Bal. |

A small piece of material is taken from the original material for metallographic testing. The metallographic specimen was subjected to 6 rounds of rough polishing and 1 round of final polishing. The order of coarsely polished sandpaper was 800#, 1000#, 1500#, 2000#, 3000# to 5000#, and the final polishing process was performed using NonDry suspension and polishing cloth for 3–5 min. Finally, the specimen is immersed in the corrosion solution for 1–5 s. The composition of the corrosion solution is 80 mL hydrochloric acid +40 mL methanol +40 g copper chloride.

The metallographic structure of the virgin material of GH4169 alloy is shown in Figure 1. In low magnification microscope photos (Figure 1a), the twin structure is marked by the red arrows, and the local area of grain refinement is marked by blue arrows. It can be observed that the grain size distribution of the material is not very uniform. Large grains have diameters of more than 50 μm, while fine grains have diameters of 10–20 μm, and fine grains are mainly concentrated in local areas. At high magnification, inclusions in the material can be seen, the size of which is 1–10 μm, mainly distributed in the grain interior.

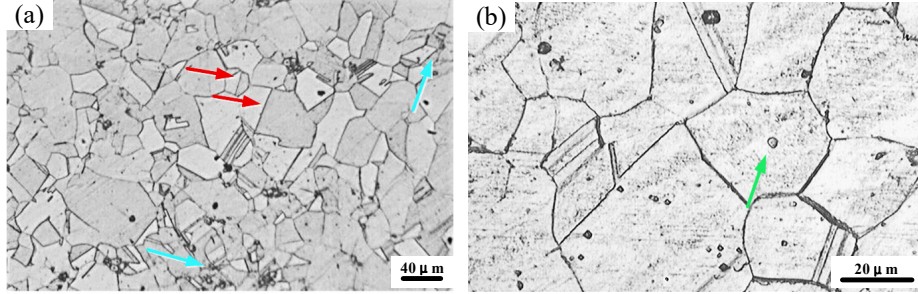

**Figure 1.** Metallographic structure of GH4169: (**a**) at low magnification; (**b**) at high magnification.

## 2.2. Specimens and Test

In this paper, three kinds of notched and a kind of unnotched plate specimens were designed. Notches in all notched specimens are of the same depth but with different radii, R = 1 mm, 5 mm, and 20 mm, respectively. The detailed dimensions of tensile and creep specimens are shown in Figures 2 and 3. The only difference between the tensile and creep specimens is the presence or absence of convex plates. The strain of the tensile specimen was measured using a laser-type non-contact extensor, so tensile specimens have no convex plates. The strain of the creep specimen was measured with a clamp-type extensometer, which needed to be clamped on the convex plates at both ends.

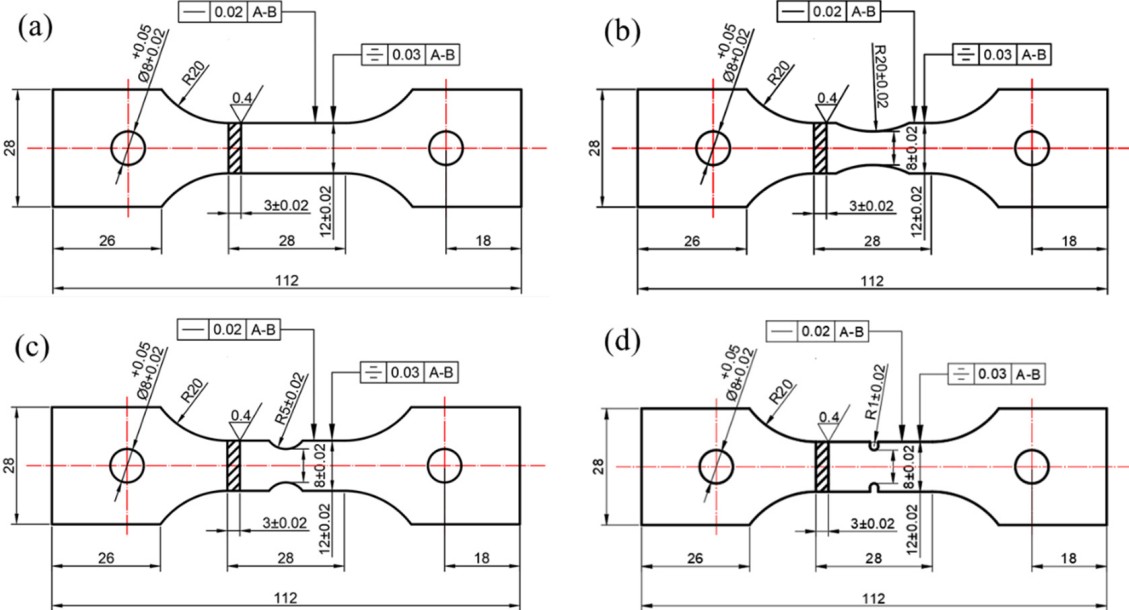

**Figure 2.** Tensile specimens: (**a**) Unnotched specimen; (**b**) Notched specimen (R = 20 mm); (**c**) Notched specimen (R = 5 mm); (**d**) Notched specimen (R = 1 mm).

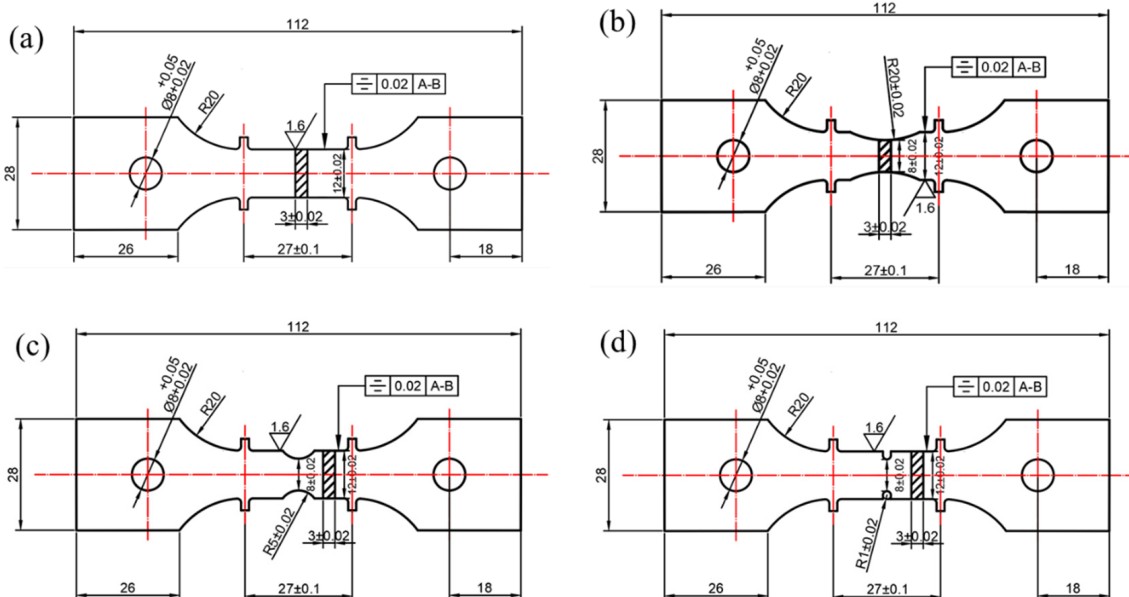

**Figure 3.** Creep specimens: (**a**) Unnotched specimen; (**b**) Notched specimen (R = 20 mm); (**c**) Notched specimen (R = 5 mm); (**d**) Notched specimen (R = 1 mm).

Tensile and creep tests were carried out in air at 650 °C. Prior to applying load, all specimens were incubated at the test temperature for 30–60 min to ensure the uniform temperature inside the specimens. Each kind of creep specimen was tested for 5–6 loads, one for each load.

### 2.3. Experimental Results

The tensile nominal stress vs. strain curves of unnotched and notched specimens are shown in Figure 4. The elastic modulus and ultimate strength of the notched plate are both higher than those of the unnotched plate, and the rupture strain is smaller than that of the unnotched plate. For notched specimens, the ultimate tensile strength increases, and the fracture strain decreases when the notch becomes sharper.

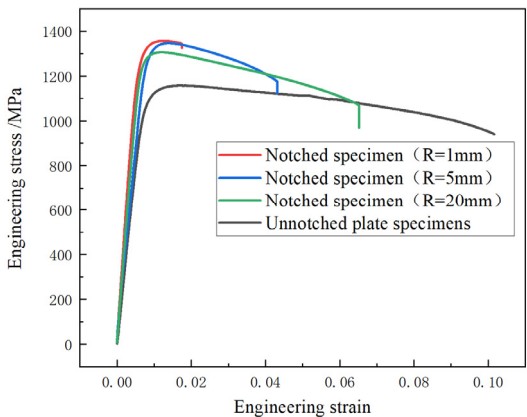

**Figure 4.** Tensile curve.

The first stage of creep of the unnotched creep test curve is not obvious, as shown in Figure 5a. But with the increase of notch stress concentration, the first stage of creep becomes more and more obvious, as shown in Figure 5b–d.

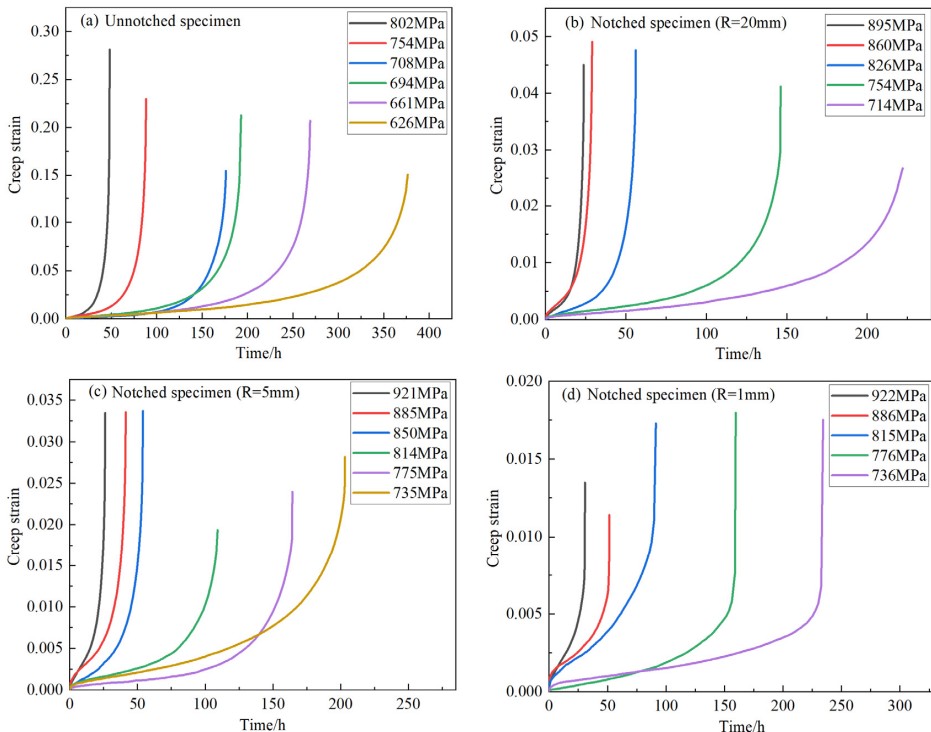

**Figure 5.** Creep curve: (**a**) Unnotched specimen; (**b**) Notched specimen (R = 20 mm); (**c**) Notched specimen (R = 5 mm); (**d**) Notched specimen (R = 1 mm).

### 3. Life Prediction Method

*3.1. Constitutive Model*

The creep analysis of notched specimens is inseparable from the finite element analysis. To understand the internal stress and strain distribution of notched specimens more clearly, it is necessary to establish the elastic-plastic coupled creep model based on tensile and creep experiments results of the material, and establish the finite element model of notched specimens by applying fixed constraints on one end of the specimens and concentrated forces on the other end.

The constitutive equation consists of the elastic-plastic constitutive coupled with creep constitutive, and the calculation process is divided into two steps: the first step is elastic-plastic loading, and the second step is creep loading. The multilinear isotropic hardening model is selected as the elastic-plastic constitutive model, and four creep constitutive models are used for comparative analysis. Since the first stage of creep experiment results (as shown in Figure 5) is not obvious, two creep constitutive models which only describe the second stage of creep are selected in this paper, and the other two creep constitutions can describe the three stages of the creep curve.

Because the notched specimen has a certain stress concentration at the minimum cross-section, the locations where the stress is concentrated will enter plasticity earlier than other locations, and the deformation of the specimen will mainly concentrate at the notch. Considering the large deformation caused by the notched position, the elastoplastic parameters of the material need to be fitted by true stress-true strain, and the conversion formulas are as follows:

$$\sigma = \sigma_e \left(1 + \varepsilon_e\right) \tag{1}$$

$$\varepsilon = \ln \left(1 + \varepsilon_e\right) \tag{2}$$

where $\sigma$ is true stress, $\varepsilon$ is true strain, $\sigma_e$ is engineering stress, and $\varepsilon_e$ is engineering strain.

3.1.1. Multilinear Isotropic Hardening Constitutive Model

The multilinear isotropic hardening behavior is described by a piece-wise linear stress-total strain curve, starting at the origin and defined by sets of positive stress and strain values, as shown in Figure 6. The first point $(\varepsilon_1, \sigma_1)$ in the model is the yield point. Theoretically, the more points selected in the curve, the more accurate the model will be. However, there can be no infinite points. In this paper, 20 points are used.

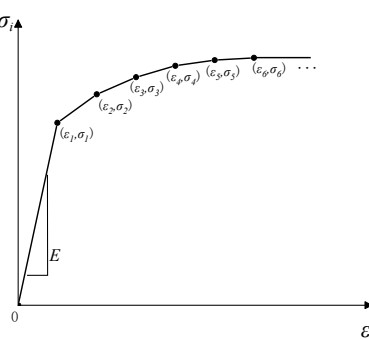

**Figure 6.** Multilinear isotropic hardening.

The elastic-plastic parameters obtained by fitting the tensile data of the unnotched specimens were input into the ANSYS for simulating the tensile properties of notched specimens. The relationship between the displacement of the gauge segment and the loading force was obtained by the finite element calculation. The results are shown in Figure 7. The experimental and calculated values of maximum loading are very close, with errors of $-1.6\%$ (R = 20 mm), $0\%$ (R = 5 mm), and $-2.2\%$ (R = 1 mm), respectively. The tensile behavior of notched specimens can be well described by the multilinear isotropic hardening constitutive.

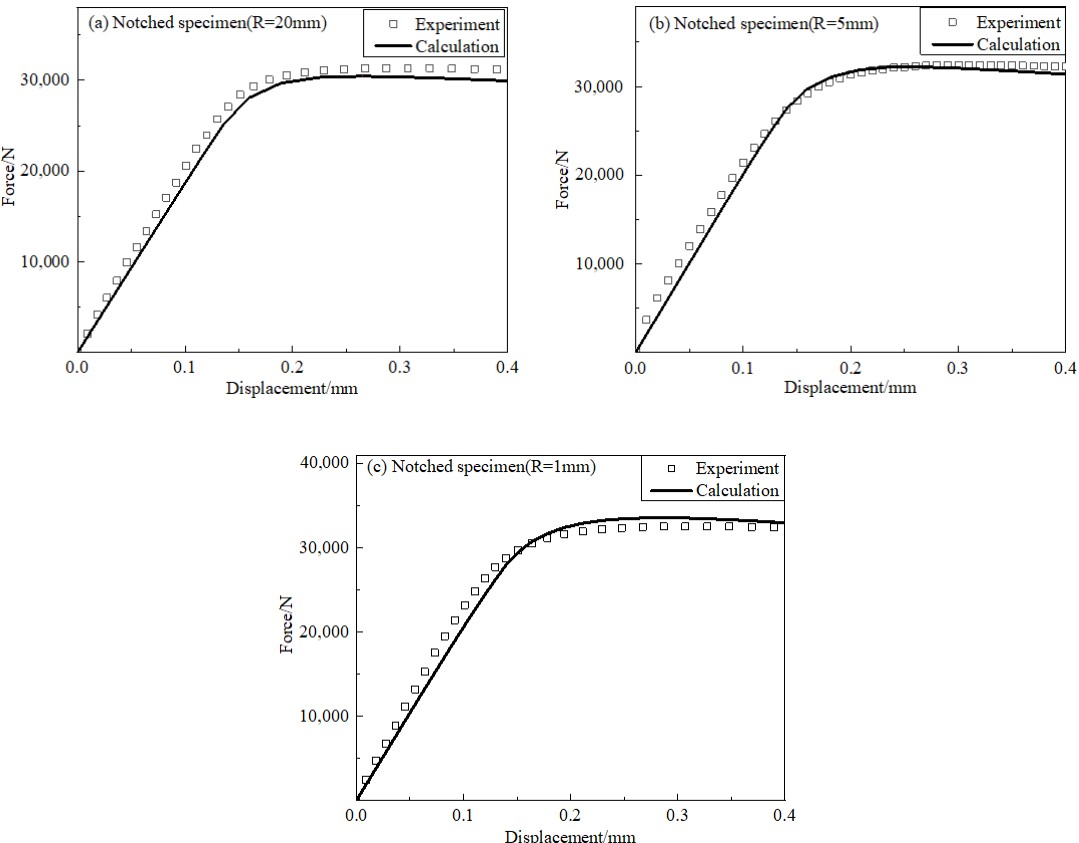

**Figure 7.** Displacement of notched specimens vs. force: (**a**) Notched specimen (R = 20 mm); (**b**) Notched specimen (R = 5 mm); (**c**) Notched specimen (R = 1 mm).

### 3.1.2. Creep Constitutive Model

Two creep constitutive models which only describe the second stage and two creep constitutive models capable of describing the three stages of creep were selected. The creep constitutive models capable of describing the whole process of creep were embedded into the finite element software ANSYS interface program usercreep f.

1. Norton law

$$\dot{\varepsilon}^c = A\sigma^n \tag{3}$$

where $A$ and $n$ are material constants.

2. Exponential form

$$\dot{\varepsilon}_c = B\exp(\sigma/d) \tag{4}$$

where $B$ and $d$ are material constants.

3. $\theta$ Projection approach

The principle of the $\theta$ Projection approach is that creep curves under uniaxial constant stress measured over a range of stresses and temperatures can be "projected" to other stress/temperature conditions and then reconstruct the complete creep curves [14]. The $\theta$ Projection approach can be written as the sum of two terms, as shown in equation (5). It is considered that the creep curve is the sum of the first stage of attenuation and the third stage of acceleration, and the second stage of relative stability is a process of the relative balance between attenuation and acceleration [15].

$$\varepsilon^c = \theta_1(1 - e^{-\theta_2 t}) + \theta_3(e^{\theta_4 t} - 1) \tag{5}$$

where $\theta_1$ and $\theta_3$ are "scaling" parameters that define the extent of the primary and tertiary creep with respect to strain. Whereas $\theta_2$ and $\theta_4$ are "rate" parameters characterizing the

curvature of the primary and tertiary creep curves [16,17]. The effect of temperature is not considered in this paper, so $\theta$ value is written as a function of stress and temperature:

$$\lg\theta_i = b_i + f_i\sigma \tag{6}$$

where $b_i, f_i$ ($i$ = 1, 2, 3, 4) are material parameters.

## 4. Ye model

The Ye model was recently developed by our research team [18]. The creep constitutive replaces the nominal stress in the traditional creep model with the normalized stress of tensile strength and can describe the whole process of creep.

$$\dot{\varepsilon}^c = \frac{\beta}{(\delta t + \varsigma)\ln(\delta t + \varsigma)} + \mu c_6 t^{c_6 - 1} \tag{7}$$

where

$$\beta = -\left(\frac{\sigma}{\sigma_{UTS}}\right)^{c_3} \exp(c_4) \tag{8}$$

$$\delta = \left(\frac{\sigma}{\sigma_{UTS}}\right)^{c_1} \exp(c_2) \tag{9}$$

$$\mu = (\sigma/\sigma_{UTS})^{c_7} \exp(c_8) \tag{10}$$

where $\sigma_{UTS}$ is ultimate tensile strength, $c_1$–$c_8$ are material constants.

The constitutive parameters are listed in Table 2, and the fitting results are shown in Figure 8. Norton law and exponential form can describe the second stage of creep well, and the $\theta$-projection method and Ye model almost fit perfectly.

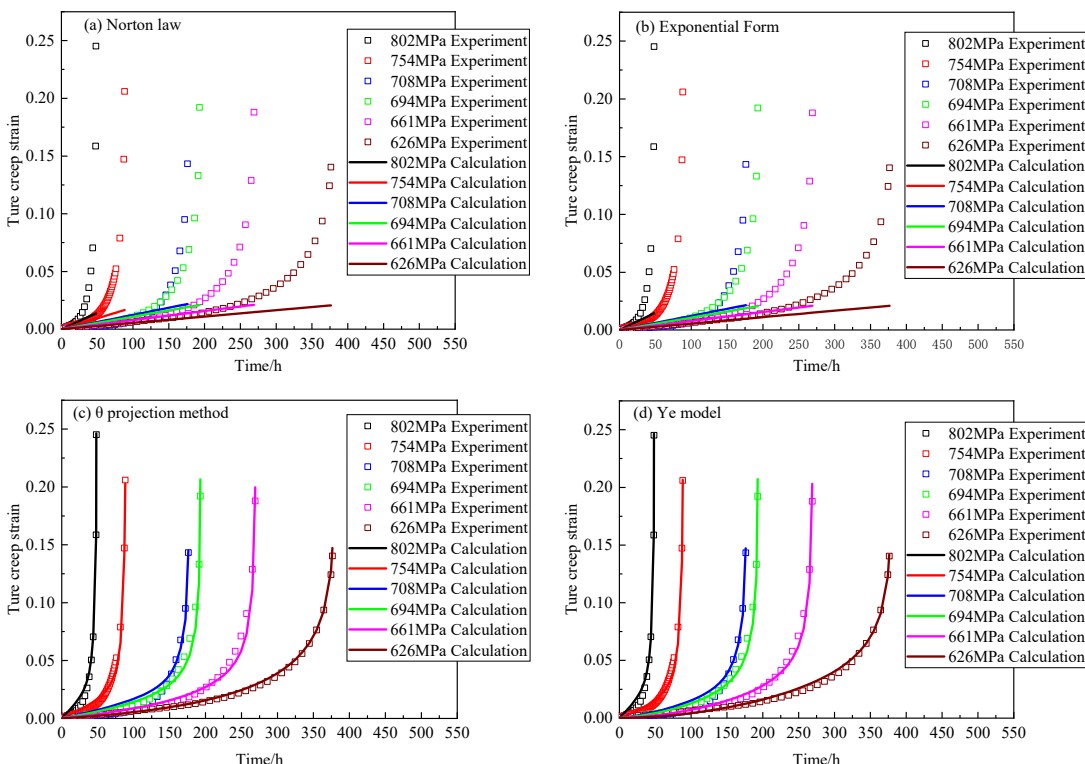

**Figure 8.** Creep constitutive models fitting results: (**a**) Norton law; (**b**) Exponential form; (**c**) $\theta$ projection method; (**d**) Ye model.

**Table 2.** Parameters of constitutive models.

| Norton law | $A$ | $n$ | | | | | | |
|---|---|---|---|---|---|---|---|---|
| | $1.62 \times 10^{-3}$ | 6.625 | | | | | | |
| Exponential form | $B$ | $d$ | | | | | | |
| | $1.41 \times 10^{-7}$ | 104.68 | | | | | | |
| $\theta$ projection method | $b_1$ | $b_2$ | $b_3$ | $b_4$ | $f_1$ | $f_2$ | $f_3$ | $f_4$ |
| | $-0.885$ | $-10.2$ | $-5.50$ | $-2.05$ | $-1.1 \times 10^{-4}$ | $9.7 \times 10^{-3}$ | $5.1 \times 10^{-3}$ | $-2.42 \times 10^{-4}$ |
| Ye model | $c_1$ | $c_2$ | $c_3$ | $c_4$ | $c_5$ | $c_6$ | $c_7$ | $c_8$ |
| | 1.955 | 6.726 | $-10.5$ | 16.74 | $-9.26$ | $1.0 \times 10^{-5}$ | 19.31 | $-9.41$ |

*3.2. Von Mises Equivalent Stress*

Since the Von Mises equivalent stress was distributed unevenly on the cross-section, the concept of AVES was introduced to characterize the Von Mises equivalent stress of the whole cross-section. The AVES is obtained by adding the equivalent stress over small increments of the area and divided by the total area as shown in Equation (11).

$$\overline{\sigma}_{VM} = \frac{1}{A} \int_0^A \sigma_{VM} dA \tag{11}$$

The curves of AVES with time are shown in Figure 9. The shape of the curves are similar to whatever the constitutive model used. The AVES of unnotched specimens increased slowly during creep, while the AVES of notched specimens first decreased rapidly and then increased slowly with time. The shape of the curves is similar to the stress-relaxation curves of the material. This is because in the initial stage of creep, the total displacement of the notched specimen varied extraordinarily little, and the stress on the notched section was higher than that on other sections. So, the stress on the notched section was redistributed. During the creep process of the notched specimen, the minimum notched cross-section began to shrink, resulting in the moderate increase of the AVES. If the small deformation finite element calculation was used rather than the large deformation calculation, the AVES of the smallest section would not increase after reaching the minimum value.

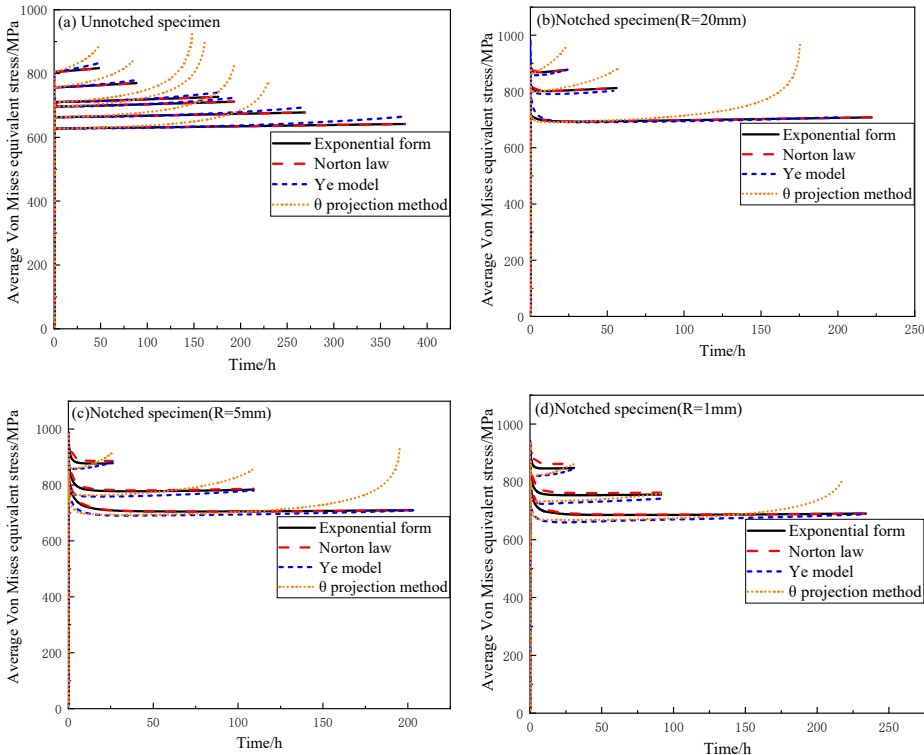

**Figure 9.** The AVES variation with time: (**a**) Unnotched specimen; (**b**) Notched specimen (R = 20 mm); (**c**) Notched specimen (R = 5 mm); (**d**) Notched specimen (R = 1 mm).

The relationship between the minimum value of AVES and the nominal stress of different notches and constitutive models is shown in Figure 10. The MAVES of the unnotched specimen is the same as the nominal stress loaded. With the increase of notch stress concentration coefficient, the difference of the MAVES calculated by different constitutive models gradually increases. The MAVES calculated by the second stage model is larger than that calculated by the whole stage model. The calculation results of the same type of creep constitutive models are very close.

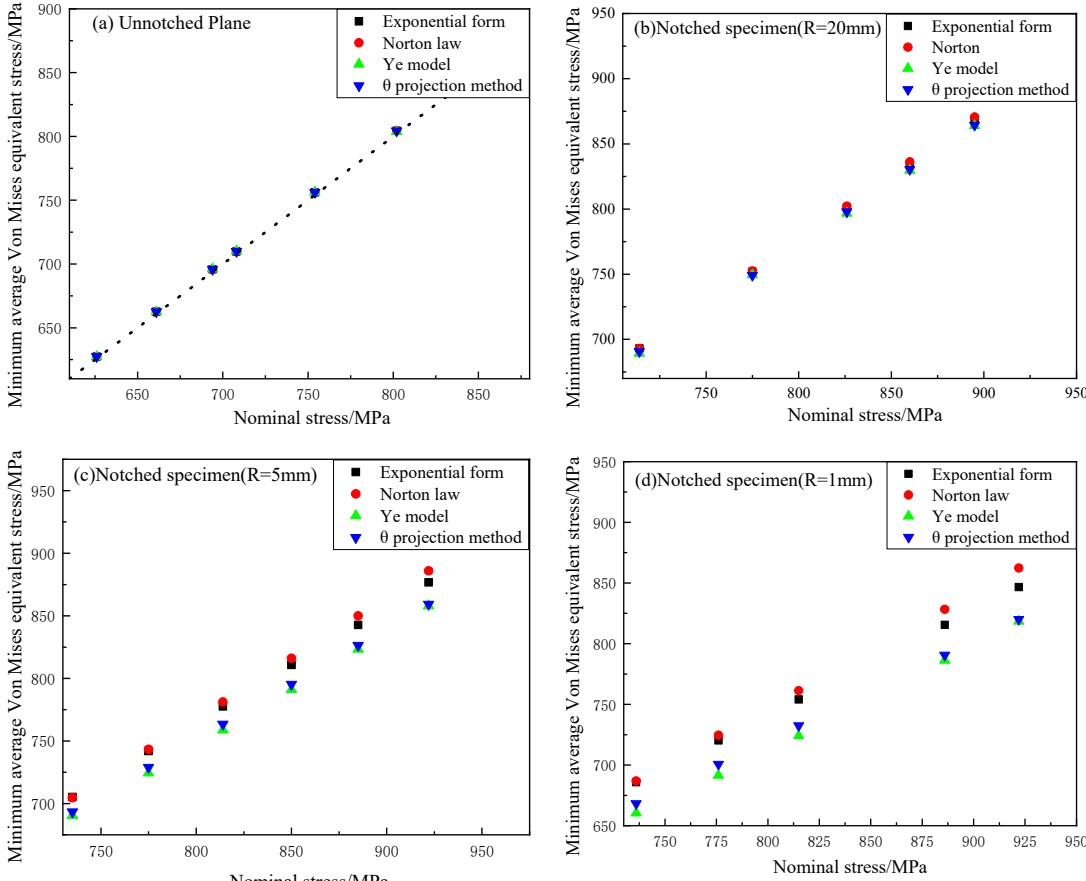

**Figure 10.** The MAVES vs. nominal stress: (**a**) Unnotched specimen; (**b**) Notched specimen (R = 20 mm); (**c**) Notched specimen (R = 5 mm); (**d**) Notched specimen (R = 1 mm).

### 3.3. Creep Life Equation

Three creep life equations are applied to evaluate notched specimen life prediction based on the MAVES. The Monkman–Grant equation [19–22] and the Larson–Miller equation [22–26], developed in the 1950s, are still widely used today. The latest creep life equation, developed by Wilshire et al. [27], has been rapidly applied to the study of creep properties of different materials [28–32].

1.    Monkman–Grant equation

Monkman and Grant [19] found a functional relationship between creep rupture life and the minimum creep rate of materials. They proposed a power law equation, containing the Arrhenius term, where the minimum creep rate is related to the creep life through the Monkman–Grant relationship as shown in Equation (12):

$$M/t_f = \dot{\varepsilon}_m^c = A'\sigma^{n^*} \exp(-Q_c/RT) \tag{12}$$

where $M$ is the Monkman–Grant constant, $A'$ and $n^*$ are material parameters, $Q_c$ are the parameters related to the activation energy, $R$ is the gas constant, and $T$ is the absolute

temperature. As there is only one loading temperature in this paper, the equation can be simplified as:

$$t_f = A'' \sigma^{n^*} \tag{13}$$

2.  Larson–Miller equation

The Larson–Miller equation expresses the life $t_f$ and temperature $T$ as the endurance thermal strength parameter $P$. Thus, the empirical relationship between the stress and the durable thermal strength parameters is established as follows:

$$\begin{aligned} P &= T(C_{LM} + \log t_f) \\ \log(\sigma) &= a_1 + a_2 P + a_3 P^2 + a_4 P^3 \end{aligned} \tag{14}$$

where $C_{LM}$ is the Larson–Miller constant and $P$ is the parameter related to temperature and life.

3.  Wilshire equation

In recent years, a research group at the University of Swansea has proposed a creep data processing method (called the Wilshire equation) based on the microscopic deformation mechanism of materials.

$$(\sigma/\sigma_{UTS}) = \exp\left\{-k_1 \left[t_f \exp(-Q_c^*/RT)\right]^u\right\} \tag{15}$$

where $\sigma_{\text{UTS}}$ is the ultimate tensile strength of the material, and $Q_c{}^*$ is the parameter related to the activation energy. $k_1$ and the exponent u are material constants. $Q_c{}^*$ is fitted by the creep test results at different temperatures. The experiments were carried out only at 650 °C, therefore, the equation can be simplified as

$$(\sigma/\sigma_{UTS}) = \exp(-k_1' t_f{}^\mu) \tag{16}$$

The fitting parameters of the three creep life equations are shown in Table 3. The fitting results are shown in Figure 11. Regardless of the creep life equation selected, there is a certain error at 708 MPa. This may be due to experimental error. The fitting degree of the regression curve to the observed value is called goodness of fit. The metric statistic of the goodness of fit is the determination coefficient, which is represented by the symbol $R^2$. The maximum value of $R^2$ is 1, and the closer $R^2$ is to 1, the better the fitting degree of the regression curve to the observed value is. Instead, the smaller the value of $R^2$, the worse the fitting degree of the regression curve to the observed value is. Consequently, it can be observed that the results fitted by the Monkman–Grant equation are worst among the three models, and the results fitted by Larson–Miller equation and Wilshire equation are similar.

**Table 3.** Parameter values of the life equation.

| Monkman–Grant equation | $A''$ | $n^*$ | | | | |
|---|---|---|---|---|---|---|
| | 1263 | −0.1153 | | | | |
| Larson–Miller equation | $a_1$ | $a_2$ | $a_3$ | $a_4$ | $C_{LM}$ | $T$ |
| | 2.585 | 11.80 | 137.82 | −5545 | 3.023 | 923.15 |
| Wilshire equation | $\sigma_{UTS}$ | $k_1'$ | $u$ | | | |
| | 1150 | 0.1336 | 0.254 | | | |

*3.4. Analysis of Prediction Results*

The MAVES calculated by all constitutive models were substituted into different life prediction equations, and the results are shown in Figures 12–14. Whichever constitutive model and life prediction method are selected, the stress rupture life prediction results of the notched specimens are almost within the double dispersion band. Because the difference of the MAVES of the same constitutive type is very small, the life prediction results of notched specimens are close.

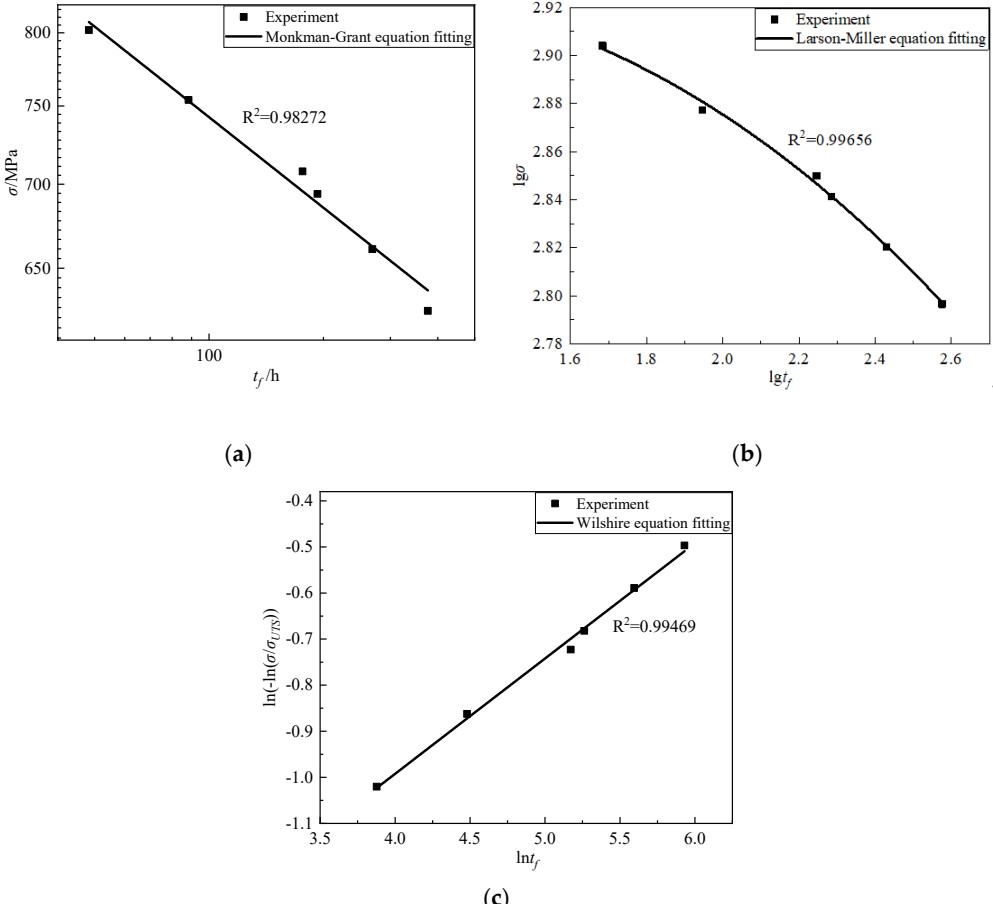

**Figure 11.** Creep life fitting results of unnotched specimens: (**a**) Monkman–Grant equation; (**b**) Larson–Miller equation; (**c**) Wilshire equation.

If the creep constitutive model capable of describing the whole process of creep and the life equation of the Wilshire equation or Larson–Miller equation are used, the prediction result will be the most ideal, which are almost within the 1.5 times dispersion band. The higher accuracy of prediction results maybe because the calculation results of the three-stage model are closer to the real load conditions.

No matter which constitutive model is selected, the results predicted by Monkman–Grant equation are all in the double dispersion band. The extrapolation ability of Monkman–Grant equation is insufficient, leading to the characteristic of small predicted values for the short-lived case as shown in Figure 13. Monkman–Grant equation has its limitations, especially in long-term data prediction based only on short-term experiments results [30], which is why the short-life prediction results in this paper are too large, as shown in Figure 13.

The prediction results are always smaller than the test results when the second-stage creep constitutive model and the Wilshire equation are selected. This life prediction method is conservative and commercial. The software usually has its own second-stage creep constitutive model. Therefore, the combination of the second-stage constitutive model and the Wilshire equation has good prospects in engineering application.

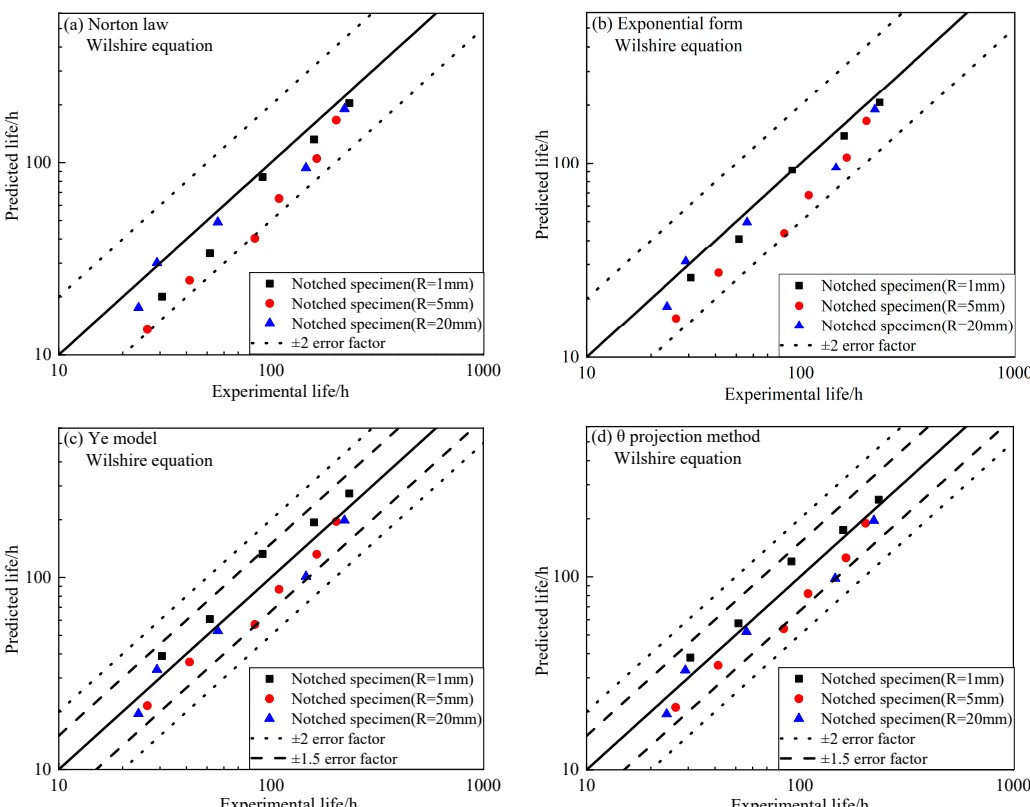

**Figure 12.** The prediction results are based on different constitutive models and the Wilshire equation: (**a**) Norton law; (**b**) Exponential form; (**c**) Ye model; (**d**) $\theta$ projection method.

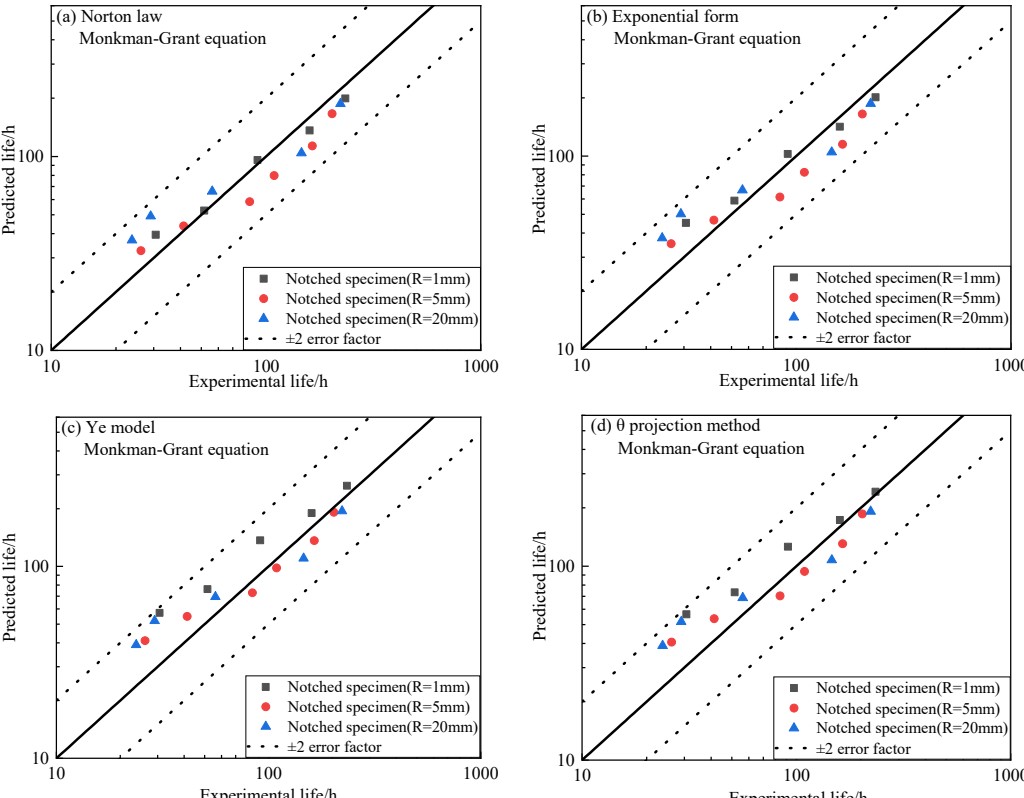

**Figure 13.** The prediction results are based on different constitutive models and Monkman–Grant equation: (**a**) Norton law; (**b**) Exponential form; (**c**) Ye model; (**d**) $\theta$ projection method.

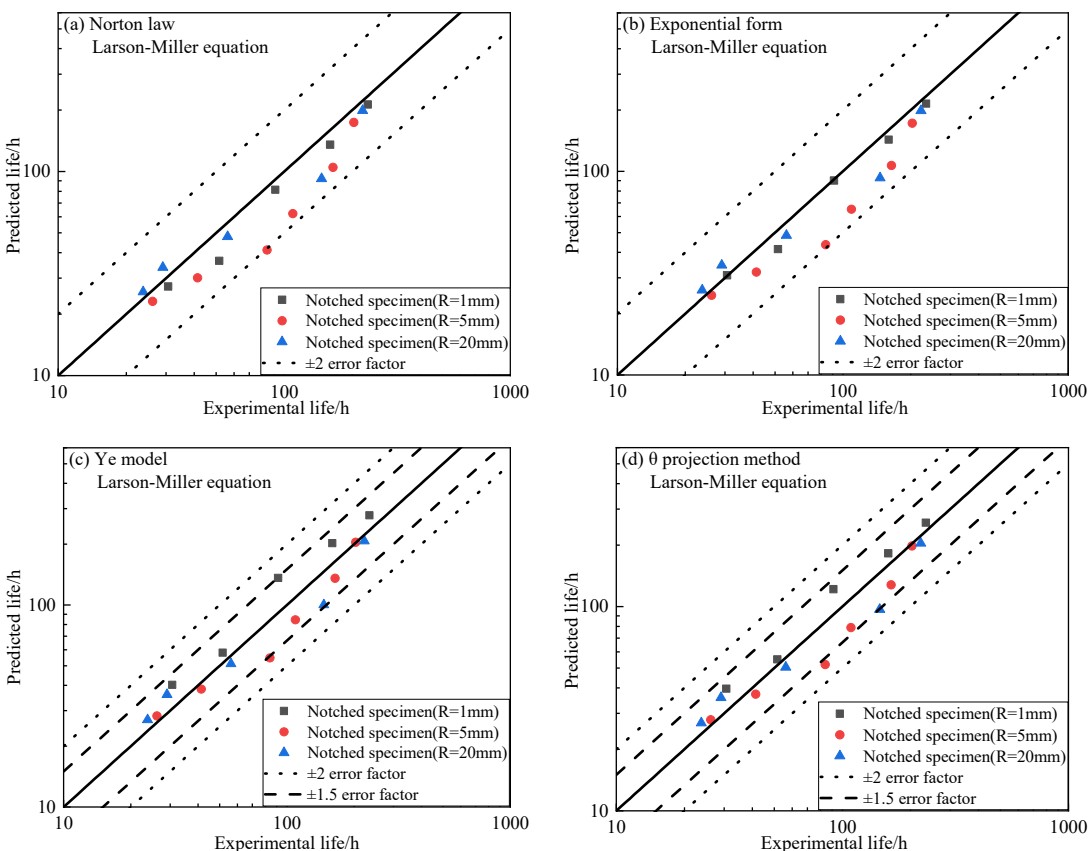

**Figure 14.** The prediction results are based on different constitutive models and Larson–Miller equation: (**a**) Norton law; (**b**) Exponential form; (**c**) Ye model; (**d**) $\theta$ projection method.

## 4. Conclusions

Based on the creep experiments on unnotched and notched specimens and finite element analysis with different constitutive models, the following conclusions have been drawn.

1. The change rule of AVES with time is that it first decreases rapidly and then increases slowly, so there is a minimum value of AVES.
2. With the increase of notch stress concentration coefficient, the MAVES calculated by the second stage model is larger than that calculated by the whole stage model.
3. Whatever the life equation or constitutive model is used, the results of notch life prediction using MAVES as the characteristic stress are within 2 times the dispersion band. If a three-stage creep constitutive model is used, the predicted results are scattered within a factor of 1.5.

**Author Contributions:** Conceptualization, D.J. and X.H. (Xianming Hu); methodology, X.H. (Xianming Hu); software, D.J.; validation, Z.Z.; formal analysis, X.J.; investigation, X.H. (Xianming Hu); resources, X.H. (Xianming Hu); data curation, Y.S.; writing—original draft preparation, D.J.; writing—review and editing, X.H. (Xuteng Hu); visualization, D.J.; supervision, Y.S.; project administration, X.H. (Xuteng Hu); funding acquisition, X.H. (Xuteng Hu). All authors have read and agreed to the published version of the manuscript.

**Funding:** This research was funded by the National Science and Technology Major Project, grant number 2017-IV-0004-0041.

**Institutional Review Board Statement:** Not applicable.

**Informed Consent Statement:** Not applicable.

**Data Availability Statement:** The data presented in this study are available on request from the corresponding author.

**Conflicts of Interest:** The authors declare no conflict of interest.

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
