# Peer review of "Stress Rupture Life Prediction Method for Notched Specimens Based on Minimum Average Von Mises Equivalent Stress"

_metals, doi:10.3390/met12010068_

Round 1

Reviewer 1 Report

referee report 
metals-1506651-peer-review-v1
Stress rupture life prediction method for notched specimens based on minimum average Von Mises equivalent stress 
Dawei Ji et al.

This manuscript reports on creep tests on notched plate specimens of the GH4169 superalloy, which is an important issue
regarding the use of this material in aero-engines. The topic is well suited for Metals. 
Overall, the present manuscript is well organized and well prepared. The present manuscript comprises 14 figures, 1 table,
and 31 references are given. 
The figures are mostly well done, but some are definitely too small (Fig. 6) or over-lettered like Fig. 8. In this case,
I recommend just to show the numbers and write in the caption that the symbols are measured data and the lines calculations.
This will make the viewing of such graphs much easier.
The experimental techniques are partly well described (microstructure), but partly not (instruments used for the stress
measurements and the procedures applied. This should be improved. 
The reference list given contains many technical problems and faults as described below.

There are several points in this manuscript which require attention prior to publication:
(1) The English requires substantial improvement. Please have a native speaker checking the manuscript. An example is line
     19 in the abstract! Another such point is line 65.
(2) Please check the spaces in the entire manuscript. There should always be spaces between quantities and units, text
    and brackets and text and citations.
(3) The abstract does not mention "superalloy", just the number. This is not useful regarding that not all readers are
     working in this field.
(4) AVES / MAVES: Abbreviations should be defined at their first use in the text.
(5) Table 1: The text (line 67) announces that GH4169 is compared to IN718, but the table provides only data for GH4169.
     And, also very important: Are these data measured by the authors on the samples investigated (how?) or are these values
     provided by the manufacturer?
(6) Fig. 1 a/b: (a) is simply too dark. Try to give a grey scale comparable to (b).
(7) What are the [J] and [M] in the paper titles for?
(8) Ref.11,13,17: incomplete citation.
(9) There is no page limits, so no need to use "et al." anywhere.
(10) Is Ref. 18 a book or a paper?
(11) Is Ref. 22 a book or a paper?

Overall, all the points mentioned above need to be corrected prior to publication.

Author Response

Point 1: The English requires substantial improvement. Please have a native speaker checking the manuscript. An example is line 19 in the abstract! Another such point is line 65.

Response 1: We apologize for the language problems in the original manuscript. The language presentation has been improved with assistance from a native English speaker with appropriate research background. The sentence in line 19 has been changed as “It was found that the average Von Mises equivalent stress on the minimum notch section first decreases and then increases with the creep time, resulting in a minimum value.”

Point 2: Please check the spaces in the entire manuscript. There should always be spaces between quantities and units, text and brackets and text and citations.

Response 2: Thank you for the suggestion. We have carefully checked the spaces in the entire manuscript. There are spaces between quantities and units, text and brackets and text and citations.

Point 3: The abstract does not mention "superalloy", just the number. This is not useful regarding that not all readers are working in this field.

Response 3: We are very sorry for this mistake. We have added “nickel-based superalloy” before “GH4169” in the abstract.

Point 4: AVES / MAVES: Abbreviations should be defined at their first use in the text.

Response 4: Thank you for the suggestion. We have added the definitions of abbreviations at their first use in the text.

Point 5: Table 1: The text (line 67) announces that GH4169 is compared to IN718, but the table provides only data for GH4169. And, also very important: Are these data measured by the authors on the samples investigated (how?) or are these values provided by the manufacturer?

Response 5: The comparison to IN718 was supported by the article and I have added the citation in the references. The values are provided by the manufacturer.

Point 6: Fig. 1 a/b: (a) is simply too dark. Try to give a grey scale comparable to (b).

Response 6: Thank you for the suggestion. The brightness of Fig. 1 (a) has been adjusted to be similar to that of (b).

Point 7: What are the [J] and [M] in the paper titles for?

Response 7: The paper is represented by [J] and the book is represented by [M]. References have been modified as required by the template.

Point 8: Ref.11,13,17: incomplete citation.

Response 8: I am very sorry for this mistake. The citations have been completed.

Point 9: There is no page limits, so no need to use "et al." anywhere.

Response 9: Thank you for the suggestion. We have deleted “et al.” in the paper.

Point 10: Is Ref. 18 a book or a paper?

Response 10: Ref. 18 is a paper.

Point 11: Is Ref. 22 a book or a paper?

Response 11: Ref. 22 is a paper.

Reviewer 2 Report

In their work, the authors raise important issues related to the evaluation of creep strength of materials intended for high temperature operation. Recently, it has been important in the case of tests of steels and alloys with austenitic matrix, intended for high-temperature operation. In the era of energy transformation, it is very important to use technologies and methods of material assessment, which enable the reduction of harmful gas emissions to the atmosphere, especially CO2. Therefore, in the opinion of the reviewer, in the current situation, research and work on the development of evaluation procedures for materials with high resistance to high temperature and stress should not be neglected. A very important element of this publication is the information on GH4169 steel. There is no doubt, then, that even rudimentary information on material degradation is widely sought after by researchers and diagnostic companies. The literature review is correct. However, the reviewer recommends reading interesting publications in a given area of service life testing, which the author should read and even mention in the introduction, which could give a broader picture of the current trends in the creep strength test.

GolaÅ„ski G., Lis A., SÅ‚ania J., ZieliÅ„ski A. (2015). Microstructural Aspect of Long Term Service of the Austenitic TP347HFG Steel. Archives of Metallurgy and Materials 60. 2901–2904

ZieliÅ„ski A., DobrzaÅ„ski J., PurzyÅ„ska H., GolaÅ„ski, G. (2015). Properties, structure and creep resistance of austenitic steel Super 304H. Materials Testing 57.  859–865

Sroka M., Zieliński A., Mikuła J. (2016). The service life of the repair welded joint of Cr-Mo / Cr-Mo-V. Archives of 561 / 5 000

Perhaps the knowledge on determining the service life contained in the above papers will contribute in the next papers to a wider understanding of the phenomenon of material degradation.

The work in terms of research is well prepared. The method of preparing the illustation also deserves praise.

Description and references are well documented.

However, the author should correct:

Standardize the magnification marking in Fig. 1

It would be useful to add the missing lines in fig 8.

The authors could add one more conclusion about the tested steel.

Author Response

Point 1: Standardize the magnification marking in Fig. 1.

Response 1: Thank you for the suggestion. The magnification marking has been standardized.

Point 2: It would be useful to add the missing lines in fig 8.

Response 2: Thank you for the suggestion. The lines in fig 8 has been modified to be clearer.

Point 3: The authors could add one more conclusion about the tested steel.

Response 3: Thank you for the suggestion. We have added one more conclusion about the tested steel.
